

# Early Pliocene deepening of the tropical Atlantic thermocline

Carolien M. H. van der Weijst[1], Josse Winkelhorst[1], Anna von der Heydt[2], Gert-Jan Reichart[1,3], Francesca Sangiorgi[1], Appy Sluijs[1]

[1]Department of Earth Sciences, Utrecht University, 3584 CB Utrecht, the Netherlands
[2]Department of Physics, Utrecht University, 3584CC Utrecht, the Netherlands
[3]NIOZ Royal Netherlands Institute for Sea Research, 1797 SZ 't Horntje, the Netherlands

*Correspondence to*: C.M.H. van der Weijst (c.m.h.vanderweijst@uu.nl)

**Abstract.**

The tropical thermocline may have played a crucial role in maintaining weaker sea surface temperature gradients during the early Pliocene and in the onset of late Pliocene northern hemisphere glaciation. Whereas the Pliocene Pacific thermocline evolution is well documented, complete records of Pliocene tropical Atlantic thermocline depths are limited to the Caribbean region. Here, we use the oxygen isotope gradient between surface to subsurface dwelling planktic foraminifera from Ocean Drilling Program Site 959 in the eastern equatorial Atlantic to track vertical changes in thermocline depth over the course of the Pliocene. This record shows that eastern equatorial Atlantic thermocline depth varied substantially during the early Pliocene, before finally deepening abruptly around 4.5 Ma to remain relatively stable until at least 2.8 Ma. Eastern equatorial Atlantic and Caribbean records are almost identical, suggesting a common control on the sudden step-wise thermocline deepening across the basin, in contrast to previous assumptions. The Pliocene evolution of the tropical Atlantic thermocline differs is remarkably from the Pacific, which is characterized by gradual basin-wide shoaling. It remains unclear what mechanisms were involved in the dichotomous thermocline evolutions. Whereas Central American Seaway closure may have shoaled the Pacific thermocline, it is not yet understood if and how this process may have deepened the Atlantic thermocline. A divergent evolution of temperatures of the source regions may explain the opposite thermocline developments observed, possibly amplified by a positive feedback loop involving tropical cyclone intensity.

## 1 Introduction

The Pliocene (5.33-2.58 Ma) is the most recent geological epoch with substantially higher greenhouse gas concentrations (De et al., 2020; Martínez-Botí et al., 2015; Stap et al., 2016) and elevated global surface temperatures (Brierley et al., 2009; Dowsett et al., 2016; McClymont et al., 2020) compared to preindustrial times. This makes it an interesting interval to study for its potential analogies with our future climate (Burke et al., 2018). Typical features of the warm Pliocene ocean are the poleward expansion of the tropical warm pools and a reduction of the zonal sea surface temperature (SST) gradient in the Pacific (Brierley et al., 2009; Dekens et al., 2008; Fedorov et al., 2013, 2015; Medina-Elizalde et al., 2008; Wara et al., 2005).



The magnitude of these gradients, however, is topic of ongoing debate due to proxy limitations and contrasting modelling results (Haywood et al., n.d.; Tierney et al., 2019; Wycech et al., 2020; Zhang et al., 2014).

Another key feature of the Pliocene ocean is the deep tropical Pacific thermocline that gradually shoaled towards the end of the Pliocene (Dekens et al., 2007; Ford et al., 2012, 2015; Steph et al., 2010), while the meridional and zonal SST gradient steepened (Fedorov et al., 2015). It has been suggested that this shoaling reached a critical threshold around 3 Ma and played an important role in the onset of northern hemisphere glaciation (Fedorov et al., 2006; Philander and Fedorov, 2003; Steph et al., 2010). Moreover, because there is general coherence between Pliocene changes in meridional, zonal, and vertical (i.e. thermocline depth) temperature gradients in the Pacific, Fedorov et al. (2015) suggested that these temperature gradients are somehow mechanistically linked. They argue that the tropical thermocline plays a crucial role in the ocean heat budget, as heat that is gained in the tropics should be balanced by heat loss at high latitudes, i.e., when more heat is gained by a tropical ocean with a shallow thermocline, more heat will be released to the atmosphere at high latitudes (Boccaletti et al., 2004; Fedorov et al., 2006; Philander and Fedorov, 2003). However, the link between ocean SST gradients, tropical thermocline depth, and ocean heat transport may be different in the Atlantic Ocean because the asymmetric geometry of the Atlantic basin leads to a different pattern of tropical thermocline ventilation (Harper, 2000). Also, at least in the modern ocean, Atlantic Meridional Overturning Circulation (AMOC) leads to northward ocean heat transport in both hemispheres, in contrast to the Pacific Ocean (Forget and Ferreira, 2019).

Interestingly, while the thermocline shoaled across the tropical Pacific, $\delta^{18}O$ and Mg/Ca records of planktic foraminifera show that the Caribbean thermocline deepened during the early Pliocene (Steph et al., 2010). This was interpreted by Steph et al. (2010) as a local phenomenon: the early Pliocene closure of the Central American Seaway (CAS) would have limited the inflow of relatively cool and fresh Pacific subsurface waters, allowing warming of the Caribbean thermocline. This hypothesis is in line with their climate model simulations, showing that while the thermocline deepened in the Caribbean, it shoaled in the Eastern Equatorial Atlantic (EEA) in response to CAS shoaling. In contrast, however, previously published surface ocean stable oxygen isotope ($\delta^{18}O$) data suggest that the thermocline deepened during the early Pliocene at EEA Ocean Drilling Program (ODP) Site 959 (Norris, 1998), suggesting that the Pliocene tropical Atlantic may have had an entirely different thermocline evolution than the Pacific.

Here, we present an EEA thermocline reconstruction from Ocean Drilling Program (ODP) Site 959, extending that of Norris, (1998), which spans the latest Miocene and almost the entire Pliocene (5.7-2.8 Ma). Thermocline depth is tracked with $\Delta\delta^{18}O_{(surface-subsurface)}$, a qualitative proxy based on vertical habitat differentiation of species of planktonic foraminifera (Ravelo et al., 1990; Williams and Healy-Williams, 1980). We use our data to investigate whether the early Pliocene Caribbean thermocline shoaling was a local phenomenon or occurred across the entire low-latitude Atlantic. Furthermore, we compare



Pliocene tropical Atlantic and Pacific thermocline evolutions and discuss what mechanisms may have been involved in shaping

them.

## 2 Methods

### 2.1 Site and modern oceanographic setting

We used sediments recovered at ODP Site 959 during Leg 159 in the Gulf of Guinea at ~160 km offshore Ghana and Ivory

Coast (3.62°N, 2.73°W; 2090 m depth; Mascle et al., 1996, Figure 1). At present, this region is characterized by a shallow

thermocline compared to the west Atlantic (Figure 2). Surface ocean circulation at Site 959 is characterized by the eastward

flowing geostrophic Guinea Current that originates from the North Equatorial Counter Current and the Canary Current (Figure

1). The thermal structure of the local water column varies seasonally in response to monsoon-induced changes in upwelling

intensity (Figure S1). In West Africa, the intertropical convergence zone (ITCZ) shifts from 5-10°N in February to 15-20°N

in August, which forces strong southwesterly monsoonal winds over the Gulf of Guinea in boreal summer (Gu and Adler,

2004). A major upwelling event occurs along the northern coast in summer, which is likely forced by a combination of local

wind-stress, wind-induced eastward propagating Kelvin waves and intensification of the Guinea Current (Djakouré et al., 2017;

Verstraete, 1992), which together raise the thermocline (Figure S1). A shorter and weaker coastal upwelling event typically

occurs in winter (Wiafe and Nyadjro, 2015). South of Site 959, upwelling occurs at the equator in response to equatorial

divergence and along the African west coast due to persistent Ekman-pumping (Bakun and Nelson, 1991; Figure 1).


### 2.2 Thermocline depth reconstructions

The thermocline is typically defined as the depth at which the vertical temperature change is at its steepest. This depth can be

approximated with the 20°C isotherm in the modern ocean, which is typically done in modelling studies, but different isotherms

may be more appropriate in warmer climates (e.g. Von der Heydt and Dijkstra, 2011). Here, we qualitatively assess changes

in thermocline depth with the stable oxygen isotopic offset ($\Delta\delta^{18}O_{(surface-subsurface)}$, hereafter $\Delta\delta^{18}O$) between the surface ocean

dwelling foraminifera *G. ruber* (late Pliocene) or *Globigerinoides sacculifer* (early Pliocene; data from Norris, 1998), and the

deeper dwelling *N. dutertrei* (Figure 3). This method builds on the concept that depth habitat differentiation among

foraminiferal species can be exploited to reconstruct the thermal structure of the water column (e.g. Birch et al., 2013; Farmer

et al., 2007; Ravelo and Fairbanks, 1992; Steinhardt et al., 2015b; Steph et al., 2009; Williams and Healy-Williams, 1980),

and has previously been used by e.g. Bolton et al., 2013; Steph et al., 2010; Wara et al., 2005. Specifically, a deep thermocline

is marked by a relatively small $\Delta\delta^{18}O$ on account of the relatively small temperature gradient between the calcification depth

of surface-dwelling foraminifera, i.e. *G. ruber* or *G. sacculifer* in this study, and the deeper dwelling *N. dutertrei*. Conversely,

a larger $\Delta\delta^{18}O$ indicates a shallower thermocline.

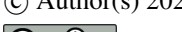



The δ[18]O in calcite depends not only on temperature, but also on the δ[18]O of the water (δw) it is precipitated in. Assuming the
modern tropical Atlantic δw-S slope of 0.11‰/salinity unit (Craig and Gordon, 1965), the upper ocean vertical salinity gradient
of 1.25 in the Gulf of Guinea (Figure S1; 0-60 m) would corresponds to a 0.14‰ δ[18]O gradient. This is much smaller than the
1.9‰ δ[18]O gradient caused by the 7.5°C temperature change over the same interval (~0.25‰/°C; Shackleton, 1974). We
therefore assume that, although the obtained values are also to some degree affected by variations in salinity, temperature
provides the predominant control on the vertical Δδ[18]O gradient.


Several other confounding factors limit the use of Δδ[18]O as a quantitative proxy for thermocline depth. These include vertical
migration of foraminifera in the water column (e.g. Steph et al., 2009), spatiotemporal variability in the mass and/or
preservation of the high-δ[18]O secondary calcite crust that *N. dutertrei* precipitates at the end of its lifecycle (Jonkers et al.,
2012; Steinhardt et al., 2015a), and the potential impact of differences in sea water carbon chemistry (Spero et al., 1997) and
light intensity (Spero and Michael, 1987) on the precipitated calcite. Moreover, a discrepancy is observed in depth habitat
ranges of planktonic species based on tows versus those reconstructed based on their δ[18]O values in core tops (Rebotim et al.,
2016). Crucially, however, Δδ[18]O performs well in predicting larger thermocline patterns in the tropical Atlantic (Steph et al.,
2009) and seems therefore suitable to qualitatively assess past thermocline depth changes.

In addition to our new late Pliocene data, we also use the 959 records from Norris, (1998), who used *G. sacculifer* instead of
*G. ruber,* to trace upper ocean conditions. Whereas both species reflect surface ocean conditions in their δ[18]O (Anand et al.
2003), they commonly have a slight isotopic offset. To compensate for this, we apply a -0.33‰ correction to *G. sacculifer,*
based on Steph et al. (2009). This offset is nearly identical to the average offset here observed in our 21 late Pliocene paired
measurements (-0.35‰ +/- 0.31‰ 1σ). This is slightly larger than the 0.17‰ offset in the Anand et al., (2003) dataset.


### 2.3 Stable isotope analysis

Samples were taken at 5-20 cm intervals from cores 959C-5H and 959C-6H between 35.77 and 48.73 revised Meters
Composite Depth (rMCD) following the splice of (Vallé et al., 2016). We combine our new records with previously published
early Pliocene data from Norris, (1998) from the 55.32 to 97.17 rMCD interval in cores 959C-7H to 959C-10H. The entire
interval consists of nannofossil/foraminifer ooze (Mascle et al., 1996). The preservation of planktonic foraminifera in the
Pliocene of 959 is generally very good; a large proportion of the tests have a "glassy" appearance, and *G. ruber* often still has
spines around its aperture (Figure S2). These features suggest that diagenetic calcite overgrowth, dissolution and
recrystallisation are minimal (Edgar et al., 2015).

For each measurement, 50-60 foraminiferal tests (*G. ruber* white s.s. and *N. dutertrei)* were picked from the 250-355 µm size
fraction. Cleaning of *G. ruber* was performed at Utrecht University and that of *N. dutertrei* at the Royal Netherlands Institute





for Sea Research (NIOZ) following slightly different protocols to remove clay and coccoliths. After gently crushing the tests between two glass plates, *G. ruber* was ultrasonically cleaned with ethanol and rinsed with distilled water. Crushed tests of *N. dutertrei* were ultrasonically cleaned three times with methanol, two times with hot (95°C) 1% NaOH/$H_2O_2$ solution and rinsed
with Milli-Q in between each step. Stable isotope analyses were performed at Utrecht University on a Thermo Finnigan GasBench-II carbonate preparation device coupled to a Thermo Finnigan Delta-V mass spectrometer. The international IAEA-CO-1 standard was measured to calibrate to the Vienna Pee Dee Belemnite (VPDB) and the inhouse NAXOS standard to correct for drift and track analytical precision, which was better than 0.1‰.

**2.4 Age model**

The Pliocene interval at Site 959 was initially dated using foraminifera and nannofossil biostratigraphy (Norris, 1998; Shin et al., 1998) and more recently updated with astronomical tuning of X-Ray Fluorescence (XRF) scan data, notably Fe records to an eccentricity, tilt and precession (ETP) curve (Vallé et al., 2016; 1.9-6.2 Ma) and tuning of benthic $\delta^{18}O$ to the LR04 stack (van der Weijst et al., in review (2020); 2.8-3.5 Ma). We use the van der Weijst et al. (in review, 2020) age model for the late
Pliocene and the Vallé et al. (2016) age model for the early Pliocene and latest Miocene interval (3.8-5.7 Ma).

**3 Results**

The $\delta^{18}O_{(surface)}$ record varies on short (<100 kyr) time-scales, but remains stable over the long term with an average of -1.7‰ (Figure 4). The $\delta^{18}O_{(subsurface)}$ record shows several major shifts as well as substantially fluctuating variance, separating four
intervals (roman numbers in Figure 4). Interval I (5,670-5,175 ka) and III (4,925-4,545 ka) are characterized by relatively high average $\delta^{18}O_{(subsurface)}$ values (0.61‰ and 0.25‰ respectively) and interval II (5,175-4,925 ka) and IV (4,545-2,800 ka) by relatively low averages (-0.61‰ and -0.70‰ respectively). During interval I, the variability of the $\delta^{18}O_{(subsurface)}$ record is of the same amplitude as in the surface record, but is superimposed on a steadily decreasing trend. The background $\delta^{18}O_{(subsurface)}$ values are relatively low and stable during interval II, but a strong positive excursion (2 data points) occurs at ~5075 ka.
Interval III is characterized by relatively high and highly variable $\delta^{18}O_{(subsurface)}$ values. For the entire duration of interval IV, $\delta^{18}O_{(subsurface)}$ remains relatively invariable, although a minor shift towards lower values occurs between ~3,200-3,150 ka. Because the $\delta^{18}O_{(surface)}$ record is relatively stable, the $\Delta\delta^{18}O_{(surface-subsurface)}$ record strongly resembles the inversed $\delta^{18}O_{(subsurface)}$ record. $\Delta\delta^{18}O$ is relatively low during intervals I and III and high during intervals II and IV, corresponding to relatively shallow thermoclines and deep thermoclines respectively.




## 4 Discussion

### 4.1 Basin-wide tropical Atlantic Pliocene thermocline changes

The 959 $\Delta\delta^{18}O_{(surface-subsurface)}$ record reflects a persistent deepening of the thermocline during the early Pliocene (Figure 4). A similar $\Delta\delta^{18}O$ evolution is observed at Site 1000 in the Caribbean (Figure 2 and Figure 5). The $\Delta\delta^{18}O$ changes are predominantly driven by the $\delta^{18}O_{(subsurface)}$ record at both sites, indicating that subsurface warming rather than surface cooling controlled the major changes in the vertical temperature gradient. Therefore, the observed changes imply that a basin-wide deepening of the tropical Atlantic thermocline occurred in the early Pliocene and persisted into the late Pliocene. The

thermocline deepening at Site 1000 was hence not a regional phenomenon as suggested by Steph et al. (2006a), but impacted the entire tropical Atlantic.

Interestingly, the early Pliocene is characterized by major swings in eastern equatorial thermocline depth, where it gradually changed from relatively shallow (interval I; ~5,670-5,175 ka), to deep (interval II; ~5,175-4,925 ka), to shallow again (interval

III; ~4,925-4,545 ka) and abruptly shifting to a relatively deep thermocline at ~4,545 ka, to remain deep for the rest of the studied interval (interval IV; ~4,545-2,800 ka). A similar evolution is observed at Site 1000, although the major deepening step is more gradual and seems to lead the abrupt shift at Site 959. It is possible that this apparent temporal offset is an artefact of age model inaccuracies at both sites. The early Pliocene age model of Site 959 is based on low-resolution tuning between Fe intensities and the Eccentricity Tilt Precession target curve (Vallé et al., 2016). In contrast, Site 1000 was dated with high-

resolution tuning of benthic $\delta^{18}O$ to the $\delta^{18}O$ records at Site 925/926 (Steph et al., 2006a). However, because the splices and age models of these target sites were revisited by Wilkens et al., (2017), the age model of Site 1000 may require some revisions.

### 4.2 Atlantic versus Pacific thermocline changes

Whereas the early Pliocene tropical thermocline deepened across the Atlantic basin, the tropical Pacific thermocline shoaled

(Ford et al., 2012, 2015; LaRiviere et al., 2012; Steph et al., 2006b, 2010). The majority of the records used in Pacific thermocline depth reconstructions are Mg/Ca-based temperature records of subsurface-dwelling foraminifera, which are generally consistent with $\Delta\delta^{18}O_{(surface-subsurface)}$ records (Ford et al. 2012). For consistency, we use $\Delta\delta^{18}O_{(surface-subsurface)}$ records from the Pacific to compare with the Atlantic $\Delta\delta^{18}O$ records: Site 806 (LaRiviere et al., 2012) in the Western Equatorial Pacific (WEP) and Site 1241 (Tiedemann et al., 2006) in the Eastern Equatorial Pacific (EEP) are compared to Site 959 and 1000

(Steph, 2005; Steph et al., 2006a) in Figure 5. Over the course of the Pliocene, the $\Delta\delta^{18}O$ change was smaller in the Pacific and the thermocline shifts were more gradual than those in the tropical Atlantic. Although changes in $\Delta\delta^{18}O$ are not necessarily proportional to vertical movement of the thermocline (Section 2.2), the early Pliocene fluctuations in the Atlantic records are much larger than the long-term shifts towards lower $\Delta\delta^{18}O$ values in the Pacific. This suggests that the thermal structure of the tropical Atlantic changed with a magnitude that is not paralleled by changes in the Pacific basin.




### 4.3 Controls on Pliocene thermocline changes

### 4.3.1 Central American Seaway closure

It has been hypothesized that closure of the CAS played a major role in the global tropical thermocline depth changes in the early Pliocene (Steph et al., 2010; Zhang et al., 2012). Salt transport to the North Atlantic increased as a consequence of
reduced inflow of relatively fresh Pacific surface waters, which in model simulations promotes the production of NADW (Lunt et al., 2007; Sepulchre et al., 2014; Steph et al., 2010; Zhang et al., 2012), although benthic isotope data suggests that the deep ocean water mass distribution did not change much in the early Pliocene (Bell et al., 2014). Steph et al., (2010) reasoned that an increased volume of NADW was associated with a greater volume of the "cold water sphere", which raised the tropical thermocline everywhere except for the Caribbean region. There, the loss of the Pacific as a source region caused the
thermocline to deepen. However, our results clearly demonstrate that the early Pliocene deepening of the tropical thermocline also occurred in the eastern tropical Atlantic basin (Figure 5), which is not reproduced in their simulations. Even if early Pliocene CAS closure promoted NADW production and/or AMOC strength, it is not clear how this would have affected the tropical thermocline depth, as both negative (Lopes dos Santos et al., 2010) and positive (Venancio et al., 2018) relationships between thermocline depth and AMOC strength have been inferred from proxy data. Moreover, modelling the effect of CAS
closure on the AMOC is complicated because in most climate model simulations for paleo time frames the spatial resolution is too low to resolve the rather complicated CAS throughflow. In addition, the evolution of the AMOC under increasing $CO_2$ is still debated even for future projections where the model resolution is generally higher (Cheng et al., 2013; Hirschi et al., 2020).

To our knowledge, only Steph et al., (2010) performed model experiments to specifically test the relationship between CAS throughflow and the Atlantic tropical thermocline. Because the mechanistic relationships between CAS closure, NADW production, AMOC strength and tropical thermocline depth are still poorly understood, it is important to further comparatively explore these relationships for the Atlantic and Pacific basins. Additionally, better constraints on the timing of CAS closure, which is currently heavily debated (Molnar, 2008; Montes et al., 2015; O'Dea et al., 2016; Sepulchre et al., 2014), may in
combination with improved age models for Sites 959 and 1000 (Figure 5), provide a chronological argument to link CAS closure to early Pliocene thermocline changes.

### 4.3.2 Temperature changes in source regions

Ford et al., (2012; 2015) suggested that along with the possible influence of gateway changes, the cooling of extratropical
source waters played an important role in the gradual shoaling of the Pliocene Pacific thermocline. During the early Pliocene,





midlatitudes SSTs were indeed generally higher than preindustrial (Brierley et al., 2009). However, the early to late Pliocene midlatitude temperature evolution was spatially highly variable (Figure 6). In the North Pacific for example, Sites 1010 and 1021 show substantial SST cooling, but Site 1208 does not. Furthermore, South Pacific Site 1125 shows relatively stable early Pliocene SSTs, whereas South Atlantic Site 1088 shows strong stepwise warming.


At present, the Pacific thermocline is sourced from mid-latitude surface waters in both the northern and southern hemisphere, whereas in the Atlantic, the thermocline is predominantly sourced from the southern hemisphere as a consequence of the asymmetric basin geometry (Harper, 2000). Assuming analogy in the Pliocene, the deepening of the Atlantic tropical thermocline may have been related to warming in the South Atlantic. Site 1088 indeed shows a sharp early Pliocene warming

(Figure 6), but assuming the current age model, this warming leads the initial excursion to deeper thermoclines (interval I to II in Figures 4-6) by ~200 kyr. Moreover, it is not clear if the short-term reversal to a shallow thermocline (interval III) coincided with temporary cooling at Site 1088 due to the resolution of the record. Contrasting temperature evolutions in the South Pacific and South Atlantic could potentially explain why the Pacific thermocline shoaled while the Atlantic thermocline deepened, but better chronologies and more SST data from midlatitudes are needed to empirically investigate this link.


### 4.3.3 Tropical cyclones and ocean mixing

Vertical mixing caused by tropical cyclones deepens the tropical thermocline (e.g. Bueti et al., 2014; Jansen et al., 2010) and it has been hypothesized that tropical cyclone activity was higher in the early Pliocene than at present, especially in the Pacific (Fedorov et al., 2010). Gradually increasing zonal and meridional SST gradients (Fedorov et al., 2015) would have promoted

stronger Walker and Hadley circulation since the early Pliocene, thereby reducing tropical cyclone activity and raising the tropical thermocline (Brierley et al., 2009; Fedorov et al., 2010). Early Pliocene tropical cyclone activity is expected to have been higher than today in the Atlantic basin, but still much lower than in the Pacific (Fedorov et al., 2010). Could a difference in cyclone activity, or a different distribution of heat (Bueti et al., 2014) have contributed to the dichotomous evolution of the tropical Atlantic and Pacific thermoclines?


By generating more Atlantic mid-latitude SST records, we can provide better boundary conditions to model the early Pliocene tropical cyclone activity and to assess their effect on the tropical thermocline depth relative to source water temperature changes. Because tropical cyclone activity and latitudinal SST gradients seem to be linked in a positive feedback mechanism (Fedorov et al., 2010), it is unlikely that the early Pliocene thermocline swings in the Atlantic (Figure 5) were caused by

changing tropical cyclone activity without the effect of changing source water temperatures.



# 5 Conclusions and outlook

The tropical thermocline underwent stepwise deepening across the Atlantic basin during the early Pliocene and remained relatively deep and stable until at least 2.8 Ma. At both Site 1000 in the Caribbean and Site 959 in the Gulf of Guinea, large $\Delta\delta^{18}O_{(surface-subsurface)}$ swings between roughly 5.2 Ma and 4.5 Ma signal major changes in tropical Atlantic thermocline depth. In the latest Miocene, the thermocline was relatively shallow. After ~5.2 Ma, abrupt changes in $\Delta\delta^{18}O$ indicate rapid thermocline deepening, followed by shoaling before it finally settled on a stable deep thermocline around ~4.5 Ma. The first deepening step seems synchronous at Site 1000 and 959, but Site 1000 leads the second deepening step by ~200 kyr. However, this apparent asynchronicity may be an artefact of inaccurate dating of either or both records, and better age constraints, or direct correlation via e.g. benthic $\delta^{18}O$, are needed to test this. In contrast, the tropical Pacific thermocline shoaled gradually between the early and late Pliocene. Whereas the magnitude of $\Delta\delta^{18}O$ does not necessarily scale linearly with thermocline depth, the magnitude of the Atlantic $\Delta\delta^{18}O$ records is much larger than that of the Pacific records, suggesting that the early Pliocene vertical thermocline movement in the Atlantic was not paralleled in the tropical Pacific.

It is not clear what mechanisms were involved in the observed deepening of the tropical Atlantic thermocline. Early Pliocene closure of the Central American Seaway (CAS) may have played a role, but this needs further testing with climate models and better chronology. Nevertheless, it is not obvious how CAS closure could have caused the dichotomous evolution of the tropical Atlantic and Pacific thermoclines. If, however, the vertical thermocline changes were caused by temperature fluctuations in source regions, i.e. mid-latitudes surface waters, regionally variable SST evolutions may explain the contrasting thermocline changes in the Atlantic and Pacific, but more SST data from mid-latitudes is needed to investigate this link. Finally, a positive feedback loop between meridional SST gradients and tropical cyclones, as described by Fedorov et al., (2010), may have amplified vertical thermocline movements.

## Data availability

The Site 959 data are available as a supplement to this paper and will be uploaded to the PANGAEA online data repository upon publication.

## Competing interests

The authors declare that they have no conflict of interest.



## Acknowledgements

We thank the International Ocean Discovery Program and the predecessors for samples and data, and Arnold van Dijk and

Wesley de Nooijer (Utrecht University) and Wim Boer (NIOZ) for analytical support. This work was carried out under the program of the Netherlands Earth System Science Centre (NESSC), financially supported by the Ministry of Education, Culture and Science (OCW).

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



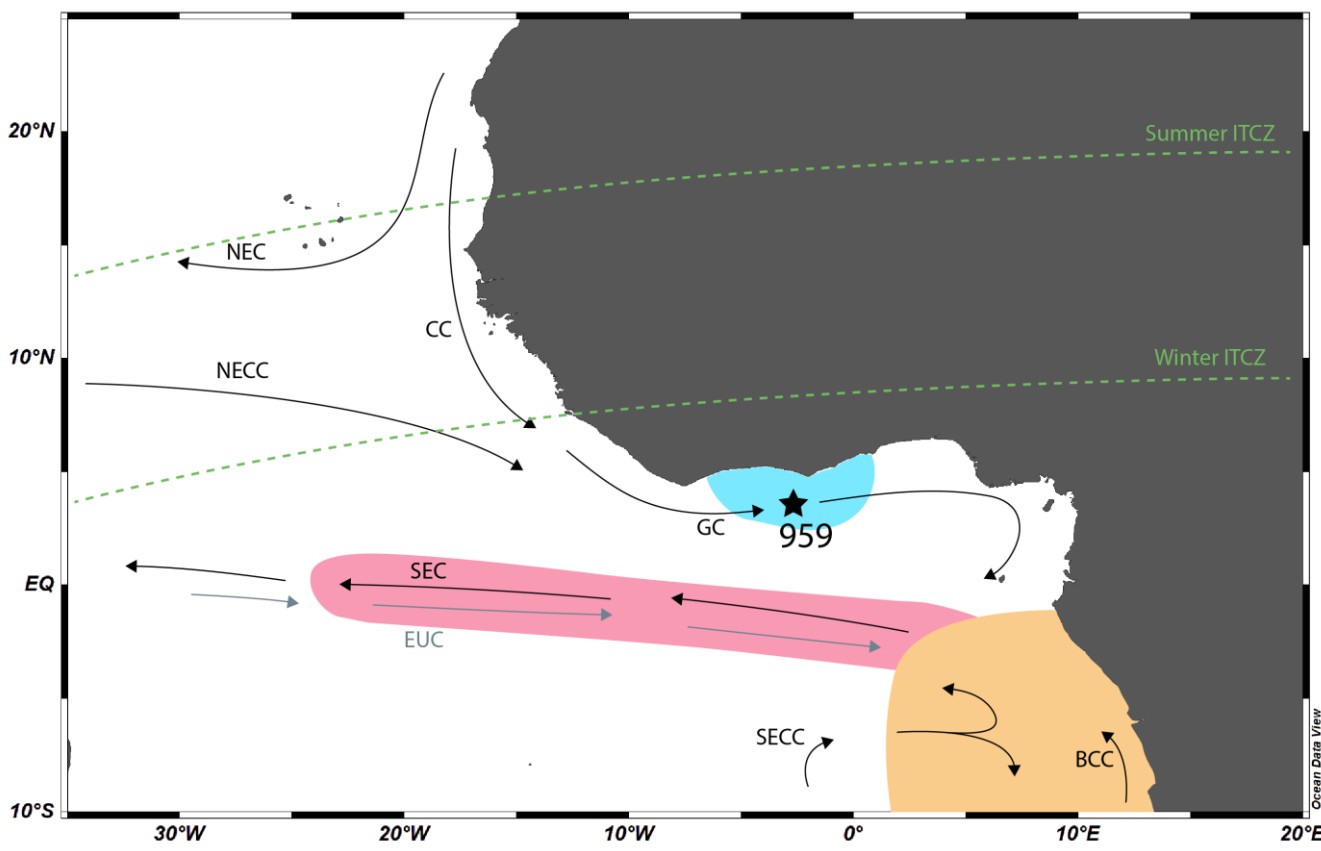

**Figure 1: General surface currents and upwelling areas in the eastern equatorial Atlantic. Blue shading: coastal upwelling in boreal summer; pink shading: equatorial upwelling; orange shading: permanent coastal upwelling. NEC: North Equatorial Current; NECC: North Equatorial Counter Current; CC: Canary Current; GC: Guinea Current; SEC: South Equatorial Current; EUC: Equatorial Undercurrent; SECC: South Equatorial Counter Current; BCC: Benguela Coastal Current. Green stippled lines mark the annual range of the Intertropical Convergence Zone (ITCZ). Figure after Norris, (1998); Wagner, (1998); Wiafe and Nyadjro (2015). Map generated with Ocean Data View (Schlitzer, 2020).**






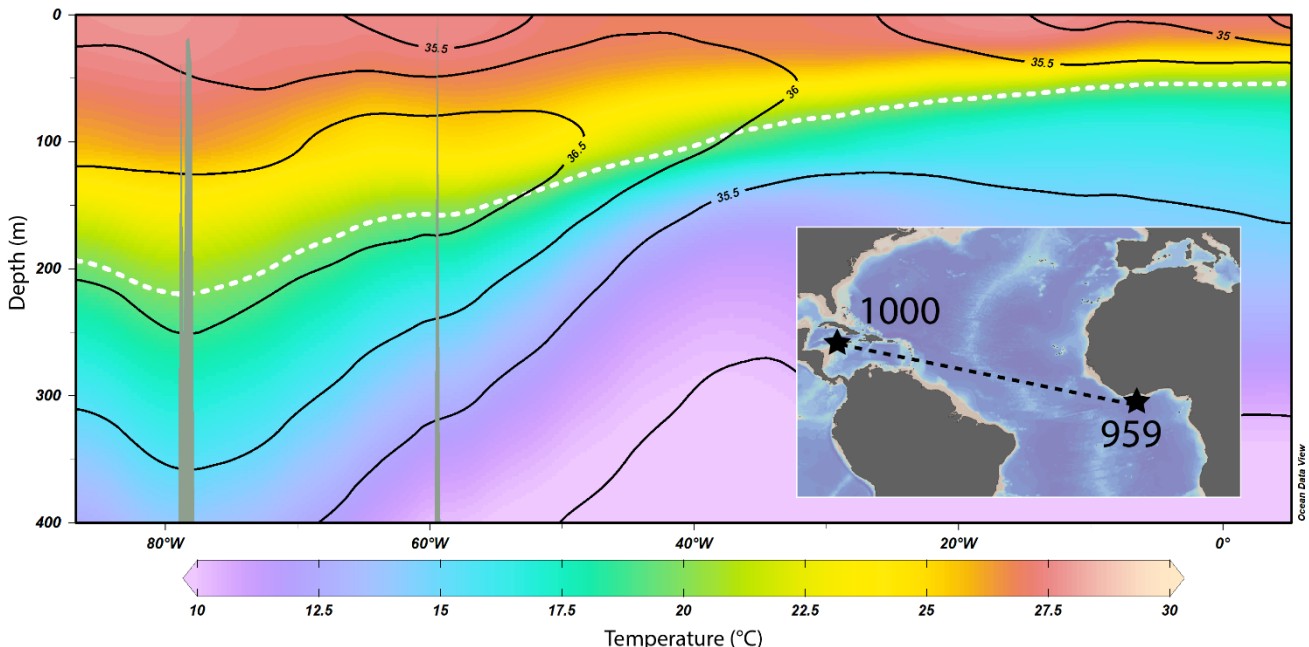

**Figure 2: Upper ocean temperature (shaded contours; Locarnini et al., 2013) and salinity (black contours; Zweng et al., 2013) profiles across the black stippled line (see map) between Site 1000 (12.74°N, 78.74°W; 2830 m depth) and Site 959 (3.62°N, 2.73°W; 2090 m depth). The white stippled line depicts the 20°C isotherm. Figure generated with Ocean Data View (Schlitzer, 2020).**




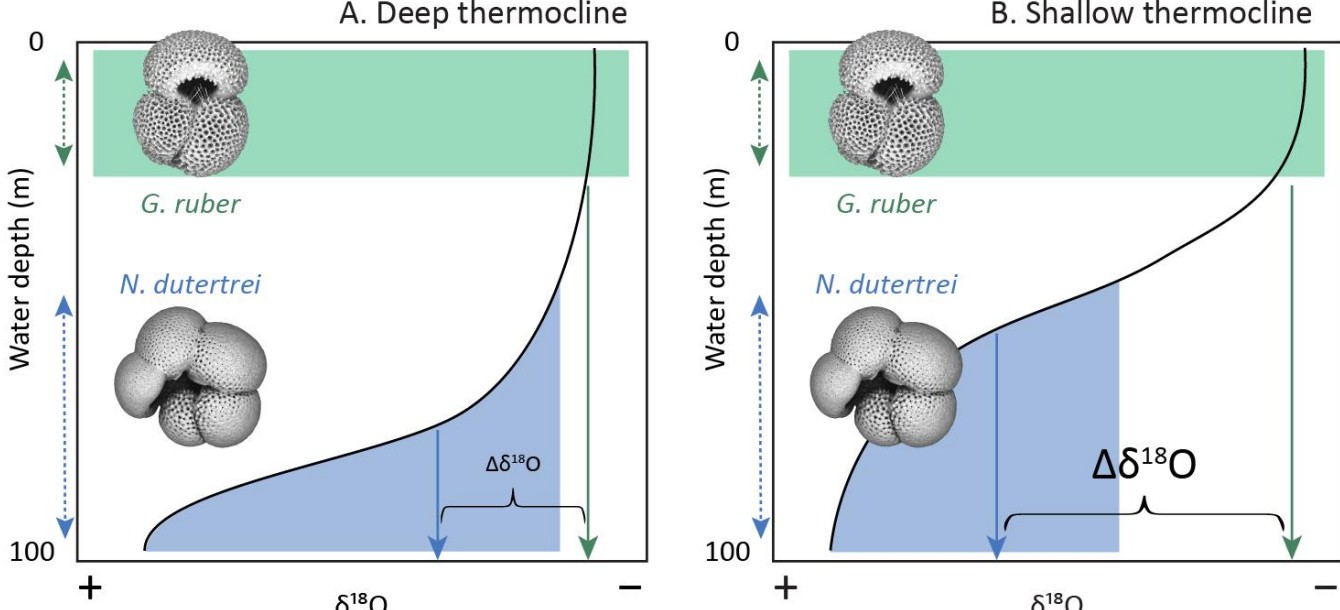

**Figure 3: Conceptual plot of vertical δ¹⁸O profiles (black; calcite equivalent of vertical temperature change), the effect of depth**
**habitat (stippled arrows) on the δ¹⁸O of exported foraminiferal tests (solid arrows), and the effect of thermocline depth on Δδ¹⁸O$_{(G.}$**
**$_{ruber - N.dutertrei)}$ (panel A vs. panel B).**







**Figure 4: Site 959 surface (*G. ruber*, light green and *G. sacculifer* (-0.33‰ adjusted, dark green) and subsurface (*N. dutertrei*, pink)**
**$\delta^{18}O$ records, as well as the vertical $\delta^{18}O$ gradient as a proxy for thermocline depth (orange). Blue shading highlights intervals with**
**a relatively deep thermocline. Roman numbers indicate the four intervals as described in the text. Data between 3.8-5.7 from Norris,**
**(1998) on the age model of Vallé et al. (2016).**



**Figure 5:** $\Delta\delta^{18}O_{(surface-subsurface)}$ **in the Western Tropical Atlantic (WTA; Site 1000), Eastern Equatorial Atlantic (EEA; Site 959), Western Equatorial Pacific (WEP; Site 806) and Eastern Equatorial Pacific (EEP; Site 1241). Data references: Site 1000: Steph, (2005) and (Steph et al., 2006a); Site 959: Norris, (1998) and this study; Site 806: (LaRiviere et al., 2012); Site 1241: (Steph et al., 2006b). Surface records were generated with** *G. ruber* **(late Pliocene at Site 959) and** *G. sacculifer* **(other sites, +0.33‰ correction to** *G. ruber***) and subsurface records with** *N. dutertrei* **(Sites 1000, 959 and 1241) and** *Globorotalia tumida* **and** *Globorotalia menardii* **(Site 806). Intervals (I-IV) are the same as in Figure 4 and Figure 6. Y-axes are evenly scaled.**



**Figure 6: Temperature evolution in potential source regions of tropical thermocline waters.** $U^{k'}_{37}$ SST records from the northern hemisphere (upper panel) and southern hemisphere (lower panel) midlatitudes. Roman numbering same as in Figure 4 and Figure 5.
