# Peer review of "Early Pliocene deepening of the tropical Atlantic thermocline"

_Climate of the Past, 2020_

## Referee Comment (RC1) · Anonymous Referee #1 · 2 Oct 2020

Van der Weijst et al. present a reconstruction of "Early Pliocene deepening of the tropical Atlantic thermocline" that may be related to the closure of Central American Seaway using stable oxygen isotopes on a mixed layer and a thermocline dwelling species of planktonic foraminifera. They conclude a sudden deepening of the thermocline around 4.5 Ma remaining for the rest of the Pliocene. Conditions before that were more variable. Three possible reasons for this deepening are presented, i.e. closing of the CAS, warming of mid-latitude SSTs, and more intense cyclones, but no hypothesis is provided for a likely scenario apart from stating that more data are needed. The location for a Pliocene reconstruction from Site 959 has a lot of potential as indeed no temperature reconstructions are available and would fill in a gap that is still existing for the Pliocene. Unfortunately, this potential is not used in this study, which I will

explain further below. In the current state I do not consider this manuscript ready for publication.

Main points: The only new data are the stable oxygen isotope values for ruber and dutertrei for the interval 3.5-3 Ma, while the main conclusions are based on data from the Norris 1998 paper. Extending the record towards 2.5 Ma would have included northern hemisphere glaciation as major event.

Why was ruber chosen here and did you not continue to use sacculifer (note that Globigerinoides has changed to Trilobatus)?

Stating the lack of temperature reconstructions from this location I was hoping to see new temperature reconstructions for Site 959, either alkenones for SST or Mg/Ca for ruber and dutertrei. This would have allowed the direct comparison with the other existing records directly for temperature. It is now argued that the present day salinity impact on d18O is not very large, but one of the main arguments for the closure of the CAS has always been the formation of the salinity gradient between the Atlantic and the Pacific, i.e. long term salinity changes of several units have taken place. Also, if periods with more or less intense upwelling did occur, this will have had an impact on salinity too; today there is a clear salinity difference between intermediate watermasses from the south and the north. So using the d18O as a pure temperature record is introducing a lot of uncertainty.

What do the stable carbon isotope data show? These show changes in productivity and/or water masse characteristics, so they may both be helping to identify changes in upwelling as well give an indication about the source waters.

Why do you choose to compare your data only with Site 1000 and not with Site 999, which has both longer and higher resolution records?

It is mentioned that the timing of the change in depth of the thermocline between 959 and 100 is off by some 200 kyr and that this may be due to uncertainties in the age

models and that this is a standing issue. First of all, I think that these orbitally-tuned age models are quite robust for the Pliocene, and if they are off it is usually not by more than 1 or 2 obliquity cycles. But secondly, if this is really the case and you have evidence for it, then provide re-tuning of these age models and present the data. Or plot the current benthic d18O of Site 1000, which is as many of the records in the area tuned to the 925, and Site 959. This record is listed as "in review", but if you want to make this part of a major explanation these data need to be included. (Lines 172-176)

There have been a lot of modelling studies dealing with the closure of the CAS, not just Steph et al., where thermocline effects related to the closure have been studied. See for example older studies like Maier-Reimer and Mikolajewicz, 1990, but also a lot of work from the Haywood, Lunt et al. groups like for example Dolan et al. 2011 (Lines 210-213). Other literature I am missing includes the basic ideas on the CAS effects like Keigwin, 1982, Haug and Tiedemann, 1998, Haug et al., 2001; thermocline reconstructions related to CAS closure like De Schepper et al., 2013; and Pliocene SST gradients in the Atlantic (Karas et al., 2017).

Minor points: Dutertrei most likely does not occur yet in the early Pliocene. This is mis-named in older papers and has probably been N. humerosa. Lines 182-189: Site 959 is the only upwelling site in the comparison, where changes always tend to be larger. So to determine if this is really an Atlantic thing, a comparison with one of the upwelling sites from the equatorial east Pacific, e.g. 847, 849, would be more clarifying. Lines 221-234: The references to all these sites are missing. Line 233: "could potentially explain". What would be your explanation then, based on this new study? Reference De et al. is incomplete. The alkenone records in figure 6 seem very low resolution to me. Was this adapted from the original records?

---

## Referee Comment (RC2) · Anonymous Referee #2 · 8 Oct 2020

In this manuscript van der Weijst et al. compare planktonic foraminifera stable isotope data (new and from other authors) to examine the evolution of the eastern equatorial Atlantic thermocline between ~2.8 and 5.6 million years ago. Previous work by Steph et al. (2005) showed a sudden deepening of Caribbean thermocline ~ 4.5 Ma, but this was interpreted to represent a local rather than regional phenomenon. The new record of van der Weijst et al however also shows a sudden stepwise deepening of the thermocline around the same time, suggesting that the changes apply to a much wider area. The authors discuss various processes that may have been responsible for these changes, but conclude that it is not clear what mechanism, or combination of mechanisms, was responsible for this deepening. The data are very interesting, I encourage the authors to take into account the comments below in a revised manuscript. 1. The

modern thermocline depth in the eastern equatorial Atlantic is rather shallow (as illustrated in figure 2). If the thermocline depth was deeper during most of the Pliocene, when did is become shallow again? And are the implications in terms of being a potential analogue for our future climate? 2. What is known about local hydrography? Is there a possibility that the locations of river mouths has shifted over time in the area. 3. In the Caribbean Steph et al. (2007) show large d18O fluctuations in G. sacculifer (and on average heavier values) between 3 and 5 Ma than at Site 959. What does this relate to? Temperature or salinity? The surface record at ODP 959 seems more similar to that of ODP 925 (Ceara Rise), although at Ceara Rise there was a gradual decrease from around 4.7 to 4.4 Ma. 4. Can there be a straightforward comparison between thermocline reconstructions of the Caribbean and eastern equatorial Atlantic, given that at least today there is a salinity maximum in the Caribbean thermocline?

Methods: The authors picked Globigerinoides ruber to reconstruct conditions in the surface ocean, and Neogloboquadrina dutertrei/ humerosa to represent subsurface water conditions. What is not clear from the text is why two different cleaning methods were applied at the two labs. Where the dutertrei specimens contaminated with organic material?

Discussion: Page 8 first paragraph: references are missing here and in the caption of Figure 6

Small issues: Abstract: Line 19 delete 'is'. Page 2 line 33: I am not sure what Haywood et al., n.d. stands for. Does this refer to the manuscript in review, or personal communications, either way it needs to be referenced properly?